# Gas Flow Measurement Method with Temperature Compensation for a Quasi-Isothermal Cavity

**Yan Shi [1], Jiaqi Chang [1], Qingzhen Zhang [1], Lijiao Liu [2], Yixuan Wang [1,*] and Zhaohui Shi [2]**

[1] School of Automation Science and Electrical Engineering, Beihang University, 37th Xueyuan Road, Beijing 100191, China; shiyan@buaa.edu.cn (Y.S.); chang_jiaqi@buaa.edu.cn (J.C.); zhangqz@buaa.edu.cn (Q.Z.)

[2] National Quality Supervision and Inspection Center of Pneumatic Products, No. 1178, Dacheng East Road, Fenghua District, Ningbo 315000, China; liulijiaoyong@163.com (L.L.); shizhaohui@live.cn (Z.S.)

* Correspondence: magic_wyx@163.com

**Abstract:** Pneumatic transmission is a technology that uses compressed air as a power source to drive and control various mechanical equipment to realize the mechanization and automation of production processes. With the development of industrial mechanization and automation, pneumatic technology, represented by pneumatic muscle, is increasingly becoming more widely used in various fields. The current standards for research are more complex for the measurement of flow without a flowmeter, some of them do not consider the influence of temperature change on flow measurement, and some of them are simplified as adiabatic or isothermal models, which are inaccurate measurement methods in actual practical application. This paper describes a method to determine flow rate by measuring the pressure change in the process of gas tank inflation. This study used the method of temperature compensation to eliminate the influence of temperature in the isothermal formula. The measurement structure was simple and the calculation was accurate, which has a certain practical significance. Based on this method, charging experiments were carried out with a gas tank that had a volume of 3 L or 5 L with or without copper wire filling, and the experimental results were used in the processed research. The temperature compensation parameters were identified with or without an isothermal environment and in different sizes of tanks. This method identified the different parameters of the 5 L tank and 3 L tank. Finally, the flow compensation was completed for the gas tank filled with copper wire. After verification, the results of the quasi-isothermal calculation formula and temperature compensation formula were close to those measured by a high-precision flow sensor in the experiment. The method introduced in this study is a novel flow calculation method that is simple in structure and accurate in calculation compared with the conventional isothermal calculation method; furthermore, it can be used in real world situations without the need for a high-precision flow sensor.

**Keywords:** quasi-isothermal tank; pneumatic flow rate; charging process; temperature compensation

## 1. Introduction

Pneumatic technology is a kind of transmission technology that uses compressed gas as a power source. Pneumatic components and pneumatic systems are widely used in the mechanization and automation of factory production processes due to the advantages of less pollution, lower cost, strong anti-interference, and convenient maintenance. The flow characteristic reflecting the relationship between pressure and flow is the most important parameter of pneumatic components. Pneumatic flow detection is widely used in the mass flow feedback of pneumatic servo control [1] and in pneumatic transportation [2–4]. Specific application scenarios include the pneumatic muscle, pneumatic spring, and air floating vibration isolation platform.

In order to design an energy-saving and low-cost pneumatic system and realize its high-precision control, it is necessary to accurately and effectively test the flow characteristics of the pneumatic components. After different test methods and several revisions for

the flow characteristics of pneumatic components proposed by some scholars, international standards ISO 6358-1-2013 [5], ISO 6358-2-2013 [6], and JIS 8390 [7] were formulated.

Among them, ISO 6358 is one of the most widely used testing methods in the world. The constant upstream pressure test method proposed by ISO 6358-1-2013 has been used as a standard to evaluate other test methods. However, due to the wide range of pressure and flow measurement, and the requirement of high measurement accuracy, this method has a low efficiency, high cost, and large air consumption; thus, it is difficult to maintain its economy and practicability. In order to overcome this, ISO 6358-2-2013 proposes an isothermal vessel venting method, which is widely used because of its low cost, high efficiency, and simple testing process. However, this alternative method still needs more in-depth research on the testing accuracy, application scope, and identification algorithm. In addition, the internal working principle of the internal temperature change and pressure dynamic characteristics of the isothermal vessel in the process of venting also need to be studied further.

In their research, Kawashima et al. proposed an alternative method to obtain sonic conductivity and a critical pressure ratio [8]. Wang et al. proposed a method to determine flow characteristics by measuring the pressure response of greenhouse emissions to the atmosphere [9]. Fujino et al. verified Wang et al.'s method under sinusoidal conditions [10]. However, the disadvantage of this method is that the serious influence of temperature change on the pneumatic flow calculation cannot be ignored. Kuroshita et al. studied a measurement method to obtain the critical pressure ratio and sonic conductivity by using small and low-noise test equipment [11]. However, in this calculation method, the default system is an isothermal system, and the influence of heat transfer and temperature change on the calculation is not considered. A test method for the flow characteristics of pneumatic components used in small equipment was proposed by Senoo et al. [12] and improved by Kuroshita et al. [13]. Thereafter, Varga et al. used the sonic conductivity and critical pressure ratio of the valve calculated according to the adiabatic model [14]. However, in practical application, the adiabatic environment is difficult to construct.

Other studies have investigated different methods, such as a new method to identify the flow characteristics of a pneumatic valve using the instantaneous polytropic index, proposed by Ye et al. [15,16]; the composite method for measuring the mass flow characteristics of high-pressure pneumatic components proposed by Gao et al. [17]; the identification of the flow characteristics of pneumatic components with sonic conductivity, critical pressure ratio, and subsonic index proposed by Liu et al. [18]; and the application of flow detection to the design of a pneumatic microfluidic chip by Yang et al. [19]. The research of these scholars did not consider the influence of temperature on flow measurement, despite the fact that temperature can never be ignored in pneumatic environments. Some scholars also did not verify their ideas through experiments, which is unrealistic.

Chabane et al. proposed a new analytical formula for mass flow rate considering the friction coefficient of pipe [20]. The formula does not explain the specific relationship between these flow parameters and the pipeline friction coefficient and is not practical if each pipe in the pipeline cannot be fully considered.

Wang et al. used a quasi-isothermal tank with temperature compensation to measure the flow characteristics of pneumatic components and introduced the pressure and temperature responses of different gas tanks and proportional valves in the discharging process [21]. This method cannot guarantee that the copper wire can create a completely isothermal environment, and the cost of completely filling the copper wire is too high. It is better to change to a small amount of copper wire without affecting the volume of the gas tank.

Based on the experience of the above scholars, this paper describes a method to determine the flow rate by measuring the pressure change in the process of gas tank charging, which solves the above shortcomings. This study also uses the method of temperature compensation to eliminate the influence of temperature deviation. The structure is simple, and the calculation is accurate, which has a certain practical significance.

## 2. Methods

### 2.1. Model Assumptions and Pneumatic Device in Charging Process

Because the relative motion of the internal parts of the flow valve and the gas flow is an interactive process, it is very difficult to determine the transient forces of the gas and other nonlinear factors. It is impossible to completely describe the change process of the gas state inside the valve and pipe in engineering. During our research, the following reasonable assumptions were made for the model:

(1) Under analytical and experimental conditions, the temperature used was room temperature and the relative pressure exceeded 5 bar; this was considered as working under normal temperature and high pressure.

(2) Under analytical and experimental conditions, there was no element to block the gas flow, and the distribution of the pressure field and temperature field in each chamber was uniform, such that the gas dynamic process was regarded as a quasi-static process.

(3) The experimental sealing performance was good; thus, gas leakage caused by a poor sealing effect was not considered.

In a pneumatic system, the air compressor is responsible for delivering high-pressure air to the system. The air enters the air tank through the flow valve, and the flow is detected by the flow sensor. In the air tank, the flow is measured by the pressure sensor and the temperature sensor. The collected data are transmitted to the upper computer through the acquisition control integrated board. At the same time, the upper computer can regulate the air compressor to output gases with different pressures through the control board. The regulating valve fills the air tank with different flow rates of gas. Here, to maintain the quasi-isothermal environment in the gas tank, the inside was filled with copper wire to increase the heat transfer.

### 2.2. Flow Measurement of Gas Tank Charging

In the equilibrium state, the basic state parameters of a simple system with only heat exchange with the outside world satisfy the following functional relationship, which is called the gas equation of state:

$$pV = mRT \tag{1}$$

where $p$ is the absolute pressure in the system, $m$ is the mass of the gas, $V$ is the volume of the gas, $R$ is the thermodynamic constant (287 N·m/(kg·K) for air), and $T$ is the thermodynamic temperature in the system.

For the differential of (1):

$$V\frac{dP}{dt} = q_m RT + mR\frac{dT}{dt} \tag{2}$$

After rearranging the formula, we obtain the following:

$$q_m = \frac{1}{RT}\left(V\frac{dp}{dt} - mR\frac{dT}{dt}\right) \tag{3}$$

If the charging process of the air tank is isothermal and the differential term of the temperature is ignored, the following equation applies:

$$q_m = \frac{V}{RT}\frac{dp}{dt} \tag{4}$$

Equations (3) and (4) show that in an isothermal chamber, the flow rate can be calculated by the change of temperature and chamber pressure. In a non-isothermal environment, the change of temperature should be considered. However, due to its own nature, temperature measurement cannot accurately measure the change of temperature, resulting in a much larger error than the current temperature value. Therefore, there is a certain deviation between the flow calculated by (3) and the actual flow, as it is usually difficult to guarantee

that a system is completely isothermal. Therefore, the influence of heat exchange in the gas tank is considered in order to analyze the deviation caused by temperature change from other angles and then accurately measure the flow through the valve.

### 2.3. Heat Exchange in a Gas Tank

Heat exchange is the process of heat transfer between two objects or parts of the same object and is caused by a temperature difference. Heat exchange is generally completed by heat conduction, heat convection, and heat radiation. The heat exchange in this system was mainly caused by heat conduction. In one system, the heat was exchanged between the filled gas and the gas tank wall. In the other system, the heat was exchanged between the gas and the copper wire placed in the gas tank in advance. Compared with air, the heat capacity of copper wire is very large. Therefore, the temperature change of copper wire was too small to be considered, and only the heat exchange of air was considered. In this system, the heat exchange included that between the outer wall of the gas tank and the external environment, and that between the air and the inner copper wire.

$$C_v m \frac{dT}{dt} = RTq_m + h \cdot S_h(T_s - T) \tag{5}$$

In Equation (5), $h \cdot S_h$ is the total heat transfer coefficient, which is the sum of the product of the respective heat transfer coefficients and the respective heat transfer areas; $T_s$ is the initial temperature when the air tank is not charging; and $C_v$ refers to specific heat at a constant volume, which is greatly affected by temperature and less so by pressure. However, the temperature increase in this study was small, and it could be approximated as 715.8 J/(kg·K) for normal temperature air.

According to (3), we obtain:

$$\frac{dT}{dt} = \frac{V\frac{dp}{dt} - q_m RT}{mR} \tag{6}$$

The relationship between temperature change, pressure change, and current temperature is deduced. Combining Equation (6) with Equation (5), the following is calculated:

$$C_v \frac{V\frac{dp}{dt} - q_m RT}{R} = RTq_m + h \cdot S_h(T_s - T) \tag{7}$$

Thus, the influence of temperature change on flow calculation can be eliminated. After arranging (7), we obtain:

$$C_v V \frac{dp}{dt} - h \cdot S_h(T_s - T) = (RT + C_v T)q_m \tag{8}$$

The overall heat transfer coefficient is calculated as follows:

$$h \cdot S_h = \frac{1}{(T_s - T)} \left( \frac{C_v V}{R} \frac{dp}{dt} - (R + C_v)q_m T \right) \tag{9}$$

### 2.4. Identification of Temperature Constant of the Gas Tank with Copper Wire

In daily life, it is unrealistic to quickly measure the heat transfer coefficient of each heat transfer component through the above formula. Due to the irregularity of the pneumatic pipeline, it is very difficult to accurately measure the heat transfer area. Therefore, in this section, the existing temperature data are used for determination by the exponential function parameter identification method. The expression for setting the overall heat transfer coefficient is as follows:

$$h \cdot S_h = K_1 e^{-K_2 t} + K_3 e^{-K_4 t} \tag{10}$$

where $K_1$, $K_2$, $K_3$, and $K_4$ are the parameters that need to be identified, and the equation satisfies the following requirements at the same time:

$$\lim_{t \to \infty} h \cdot S_h = K_1 + K_3 = h_\infty \cdot S_h \tag{11}$$

where, $h_\infty \cdot S_h$ is the overall heat transfer coefficient at the end of inflation. This method only needs to measure the overall heat transfer coefficient of the initial state, the end state, and the middle state, then the heat transfer coefficient at each point in the inflation process can be determined. By calculating and comparing the experimental data, the advantages and disadvantages of temperature constant identification become apparent.

### 2.5. Flow Measurement of the Gas Tank under Temperature Compensation

The system switched between isothermal charging and adiabatic charging. The gas tank was filled with copper wire to increase the internal heat transfer, in an attempt to create a quasi-isothermal environment; however, there was still a certain gap with an ideal isothermal environment. The isothermal formula could not be used to calculate the flow through the proportional valve, and the temperature compensation formula was used instead. The temperature of the isothermal condition was set as $T_r$. Therefore, the difference between the actual flow and the isothermal flow could be obtained by combining the isothermal equation (Equation (4)) and the actual measurement through the flow rate of the proportional valve. The flow difference was calculated under choked flow.

$$\Delta q_m = \frac{V}{RT_r} \frac{dp}{dt} - q_m \tag{12}$$

To eliminate the pressure differential and simplify the calculation, according to the charging process of the air tank, the relationship between pressure differential, flow rate, and temperature is obtained as follows:

$$\frac{dp}{dt} = \frac{R}{VC_v}(C_v T + RT)q_m + hS_h(T_s - T) \tag{13}$$

After combining Equation (13) into Equation (12) and substituting it into the flow rate, it can be concluded that:

$$\Delta q_m = \left(\frac{C_v T + RT}{C_v} - 1\right)q_m + \frac{hS_h}{T_r C_v}(T_s - T) \tag{14}$$

Thus, the actual flow value can be obtained by measuring the pressure, temperature, and other parameters before and after the valve in the charging process.

### 2.6. Parameter Identification of Pneumatic Flow

It is difficult to compensate the flow by the above formula, and in the above analysis, $hS_h$ is considered to be determined as an approximate parameter in the form of an exponential function. Therefore, the flow difference can be approximated as:

$$\Delta q_m = K_5 e^{-K_6 t} \tag{15}$$

By calculating the flow deviation at the beginning of inflation and the flow deviation at the end of inflation, the values of $K_5$ and $K_6$ can be obtained by fitting. Compared with the actual deviation and the compensation amount after parameter identification, the advantages and disadvantages of parameter identification can be measured.

### 3. Experiment

*3.1. Introduction of Experimental Device*

The experimental setup is shown in Figure 1. The gas source processing device and pressure acquisition system were designed by Festo. The flow sensor was designed by ECOSO, and the temperature acquisition component used a high-precision thermocouple. During the gas source processing, which included air compression, filtration, and pressure pre-regulation, the gas source output pressure was adjusted to 5.85 bar. After adjusting the flow valve beside the gas source outlet, the gas tank could slowly be filled; the inflation time was long enough for the system to be able to collect data at different times. In the experiment, the accuracy of the pressure sensor was 0.001 bar, the accuracy of the flow sensor was 0.1 L/min, and the accuracy of the temperature sensor was 0.1 °C.

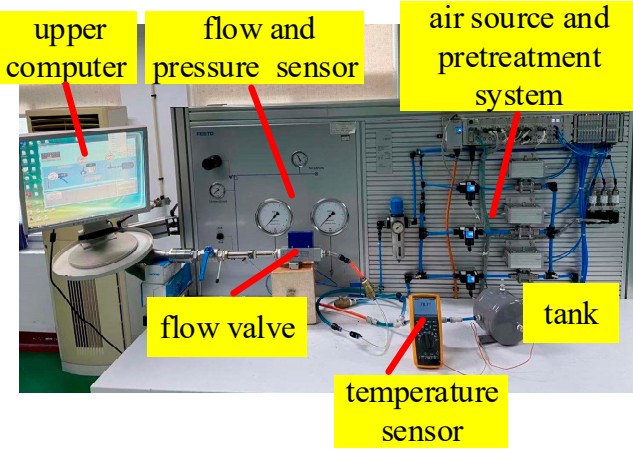

**Figure 1.** Field picture of the test device.

To maintain the quasi-isothermal environment, the gas tank was filled with copper wire, as shown in Figure 2. A 5 L gas tank and 3 L gas tank were filled according to the same proportion of volume. The copper wire was first wound into a rope with a certain gap, inserted into the opening of the air tank, and folded randomly in the air tank.

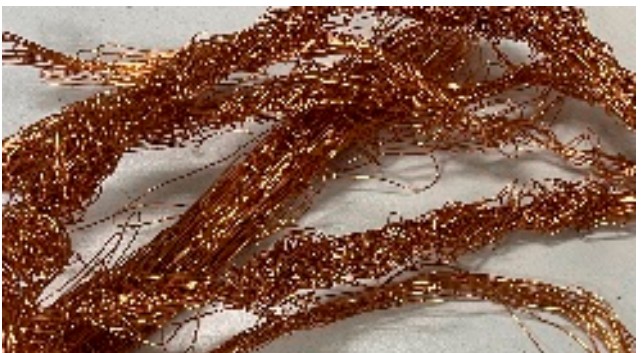

**Figure 2.** Copper wire inserted into the air tank.

*3.2. Experimental Process*

The air system was connected according to the connection method shown in Figure 1, the outlet pressure of the air source was adjusted in advance, and the flow was deflated for a period to stabilize so that the gas filled the 5 L and 3 L air tanks with winding copper wire mesh. During the experiment, all the gas flow ball valves were opened, and the gas flow was measured at the same time. When the pressures of the gas tank and gas source were balanced, the gas filling was terminated and the recording was stopped.

## 4. Results

### 4.1. Effects of Temperature and Pressure during Charging

This section describes the pressure and temperature response of the air tank inflation. According to the connection structure above, the 5 L air tank was filled with winding copper wire mesh. During the experiment, the valve of the air source was opened, and the sensors started measuring in the process of charging. At the end of inflation when the pressure of the air tank was balanced with the pressure of the air source, the experiment device stopped recording. The gas tank inflation characteristics of copper wire will be discussed in this section.

In the process of increasing inflation, it can be seen in Figure 3 that the pressure continued to rise up until the air source pressure. The temperature increased rapidly in the early stage of charging, and then increased slowly with the decrease in charging flow. In the later stage, the temperature decreased due to the increase in the temperature difference between the gas temperature and the heat exchange material. At the same time, it reached stability faster after being filled with copper wire. It can also be seen that the pressure of the gas tank with copper wire increased faster and could reach the maximum inflation pressure faster.

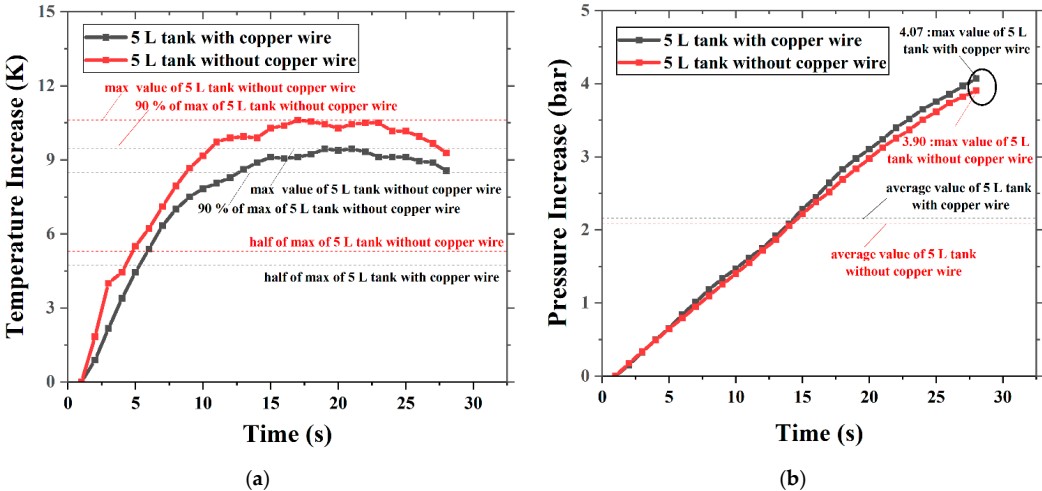

**Figure 3.** Experimental response of 5 L gas tank: (**a**) temperature response and (**b**) pressure response.

Similarly, as shown in Figure 4, the pressure and temperature responses of the 3 L gas tank with or without copper wire filling were obtained as follows.

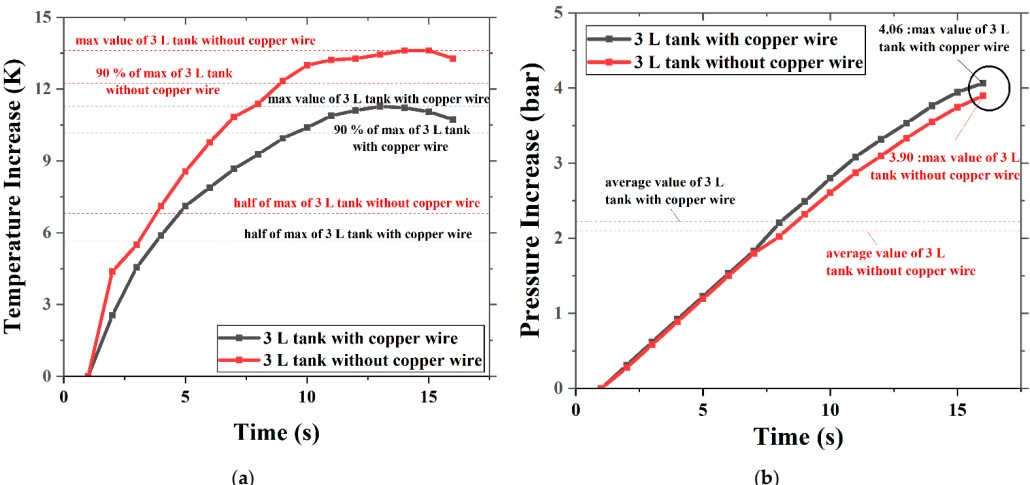

**Figure 4.** Experimental response of 3 L gas tank: (**a**) temperature response and (**b**) pressure response.

It can be seen that the temperature increased slowly and the pressure increased rapidly in the 3 L gas tank.

### 4.2. Relationship between Temperature Rise Inhibition Ratio and Copper Wire Filling

The evaluation index of the temperature rise inhibition ratio is proposed in this section to measure the quasi-isothermal effect with or without copper wire. The heating inhibition ratio formula is shown in Equation (16) as follows:

$$\beta = \frac{T - T_s}{T_s} \tag{16}$$

where $\beta$ is the heating inhibition ratio, $T_1$ is the current temperature, and $T_s$ is the initial temperature.

Figure 5 shows the variation trend of the temperature rise rate. For the 5 L gas tank, the rate increased fastest in the first 5 s and exceeded half of the maximum value in the sixth second, then slowed down due to the inhibition effect in the tenth second and entered the range of 90% in the fourteenth second. This range of 14 s to 24 s was relatively stable, reaching the maximum value of 3.147% in the 18th second; then, as the inflation process slowed down, the rate also slowed down, indicating that the benefit of copper wire filling could be maximized between 14 s and 24 s, which were the middle and late stages of charging gas. For the 3 L gas tank, the rate exceeded 50% of the maximum value in the fourth second and maintained a high growth rate. It entered the range of 90% in the 11th second, reached the maximum value of 3.78% in the 14th second, and then began to decline until the end of charging.

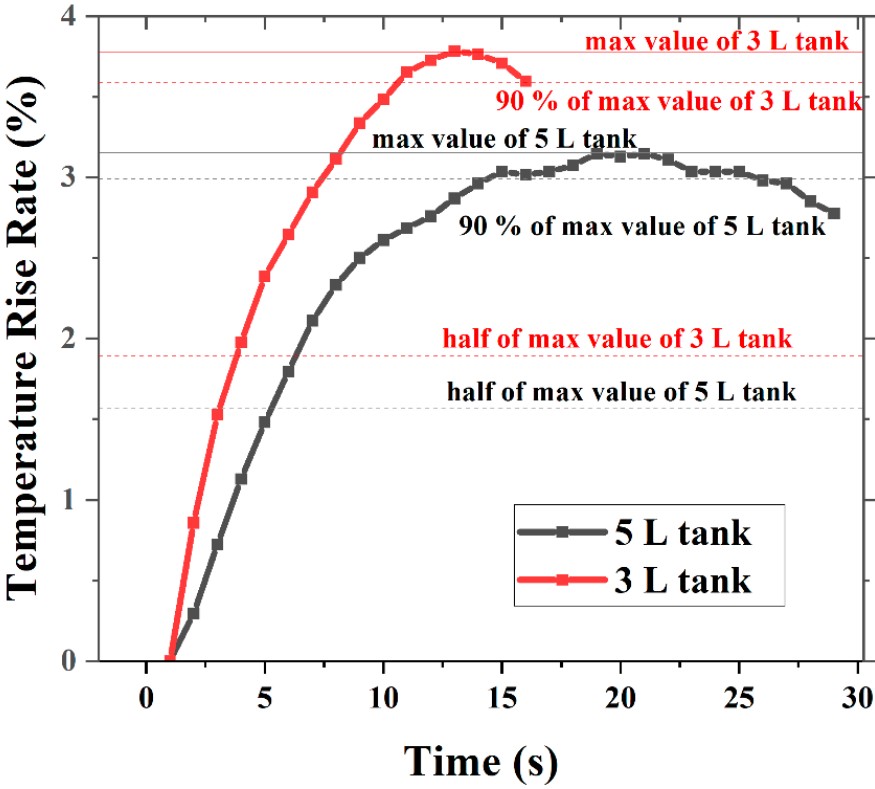

**Figure 5.** Variation trend of temperature rise rate.

It was observed that the copper wire at the beginning of inflation could be better suppressed, while the volume of the air tank did not affect the temperature suppression efficiency after the inflation process was stable.

### 4.3. Comparison of the Overall Heat Transfer Coefficient before and after Parameter Identification

As mentioned above, it is an unrealistic calculation to quickly determine the heat transfer coefficient of each heat transfer component in daily life by using the formula introduced earlier. Due to the irregularity of the pneumatic circuit, it is very difficult to accurately measure the heat transfer area. Therefore, in this section, the existing temperature data are used for determination by the exponential function parameter identification method. For 5 L and 3 L cavities with or without copper wire, the actual overall heat transfer coefficient calculated according to the previous section is shown in Figure 6.

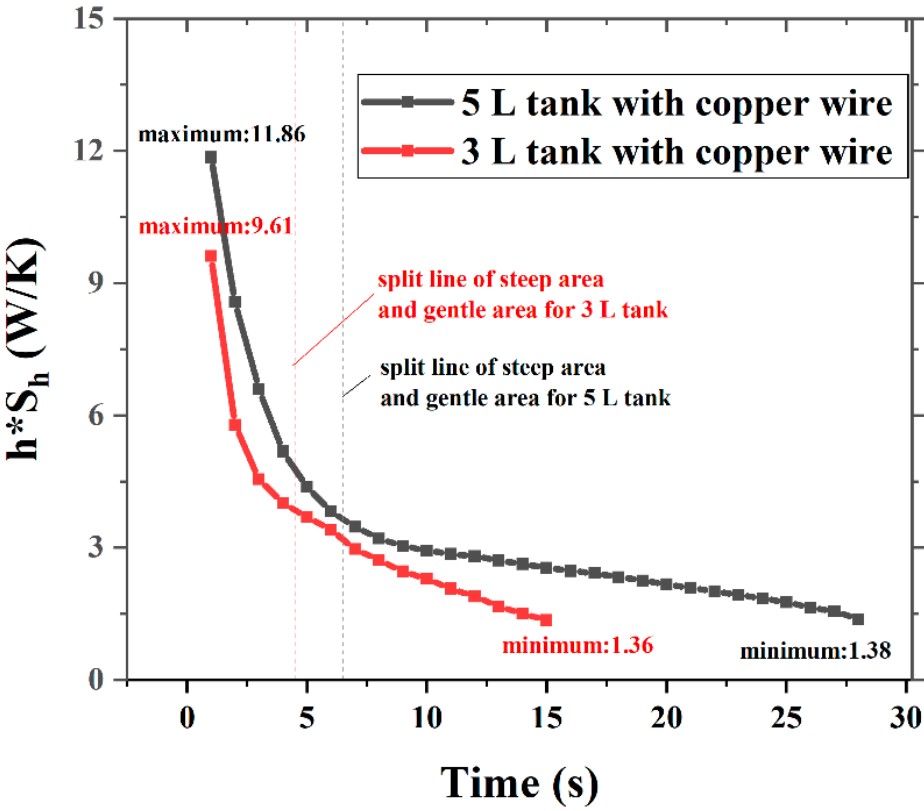

**Figure 6.** Actual value of overall heat transfer coefficient.

The actual value of the overall heat transfer coefficient showed a downward trend. At the beginning, the maximum value of the 5 L tank was 11.86 W/K. The downward slope slowed at the seventh second, and then decreased slowly until the end of recording, with a final value of 1.38 W/K. For the 3 L tank, the maximum value was 9.61 W/K before entering a sharp descent. After falling till the fifth second, the slope slowed down and then dropped slowly until the end of recording, with a final value of 1.36 W/K.

The parameters of $h \cdot S_h$ identified according to Equation (13) are shown in Table 1.

**Table 1.** Identification parameters of overall heat transfer.

| Tank | $K_1 (\mathbf{W/K})$ | $K_2 (1/s)$ | $K_3 (\mathbf{W/K})$ | $K_4 (1/s)$ |
|---|---|---|---|---|
| 5 L tank | 3.98 | 0.03168 | 13.5 | 0.5173 |
| 3 L tank | 5.903 | 0.09628 | 20.37 | 1.565 |

Figure 7 shows the comparison with the actual $h \cdot S_h$ values and the calculated $h \cdot S_h$ values.

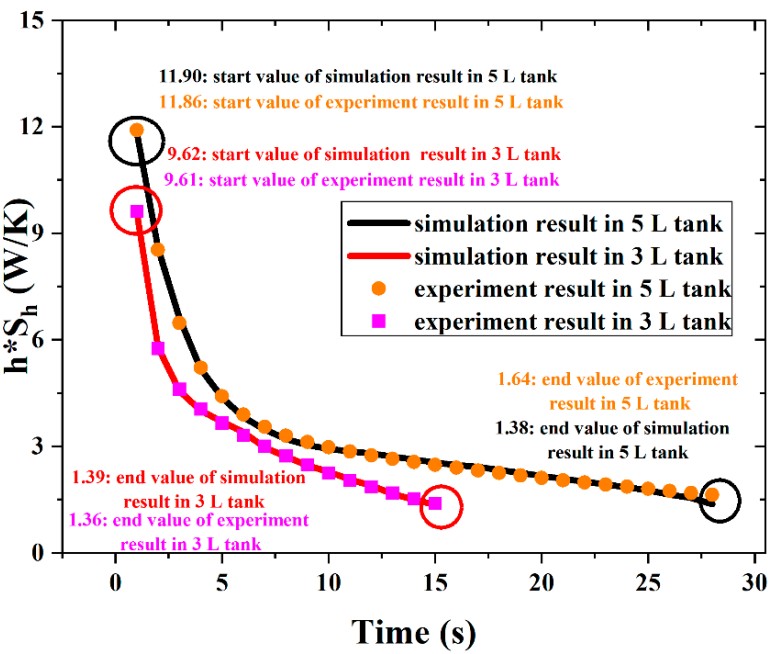

**Figure 7.** Theoretical and experimental comparison of overall heat transfer coefficient.

Comparing the theoretical and experimental results of overall heat transfer coefficient, it can be seen that the fitting effect was accurate. In the fitting effect of the 5 L tank experiment, the initial simulation value of 1.9 W/K maintained the same downward trend as the experimental value and reached 1.64 W/K, while in the fitting result of the 3 L gas tank, the initial simulation value of 9.62 W/K decreased to 1.39 W/K. The error diagram of fitting is displayed in Figure 8, showing a maximum error of 2.92% in the 5 L tank experiment at 8 s, while the minimum error was 0.22% at 11 s, and the average of the error of the whole process was only 1.63%. In the 3 L tank experiment, the maximum error was 2.69% at 6 s, the minimum error was 0.05% at the first second, and the average of the error of the whole process was 1.33%.

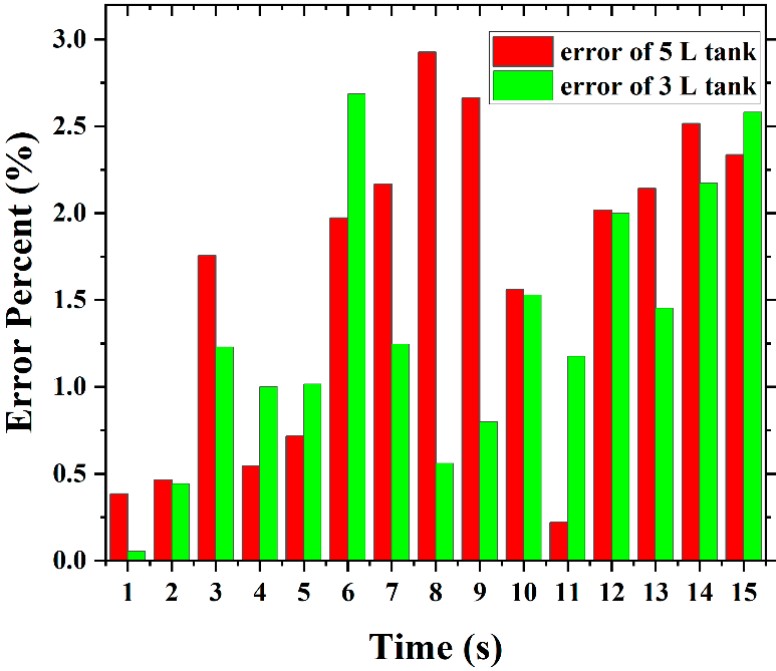

**Figure 8.** Experimental error diagram of overall heat transfer coefficient.

The indexes used to describe parameter identification include SSE, R-square, adjusted R-square, and RMSE. The goodness of fit is shown in Table 2.

**Table 2.** Goodness of fit of overall heat transfer coefficient index model.

| Tank | SSE | R-Square | Adjusted R-Square | RMSE |
|---|---|---|---|---|
| 5 L tank | 0.2116 | 0.9985 | 0.9984 | 0.09201 |
| 3 L tank | 0.02333 | 0.9996 | 0.9995 | 0.04606 |

It can be seen that under these parameters, $h \cdot S_h$ can be replaced by this model with smaller error.

*4.4. Comparation of the Fitting of Compensation Flow before and after Parameter Identification*

As mentioned above, it is difficult to compensate the flow by using Equation (15). Because $hS_h$ is determined as an approximate parameter in the form of an exponential function in the above analysis, the flow difference can be approximated as Equation (16). The charging experiment of the 5 L gas tank with copper wire is taken as situation 1 and the charging experiment of the 3 L gas tank with copper wire is taken as situation 2. The actual flow and calculated flow before parameter identification are shown in Figure 9.

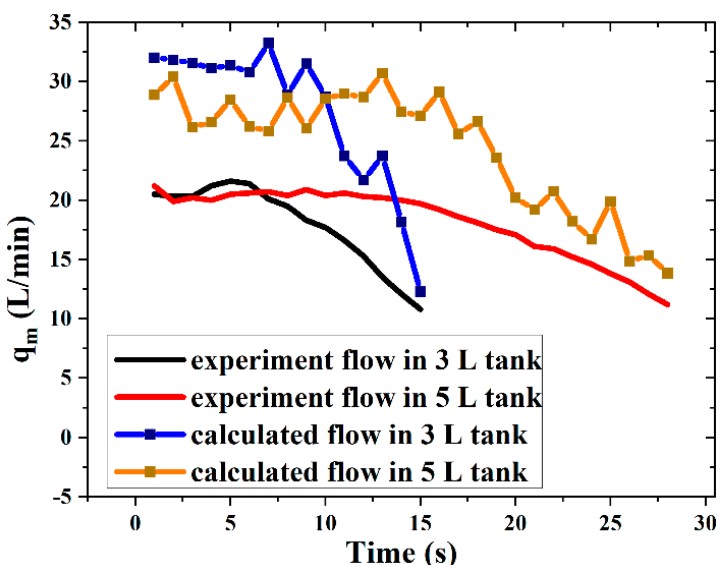

**Figure 9.** Comparison of experimental flow and calculated flow.

The parameters of $\Delta q_m$ were identified according to Equation (16), and the parameters are shown in the Table 3.

**Table 3.** Identification parameters of flow compensation.

| Tank | $K_5$ (L/min) | $K_6$ (1/s) |
|---|---|---|
| 5 L tank | 9.019 | 0.02686 |
| 3 L tank | 13.09 | 0.04287 |

By calculating the flow deviation at the beginning of inflation and the flow deviation at the end of inflation, the values of $K_5$ and $K_6$ can be obtained by fitting. Compared with the actual deviation and the compensation amount after parameter identification, the advantages and disadvantages of parameter identification can be measured. The difference and parameter identification results are shown in Figure 10.

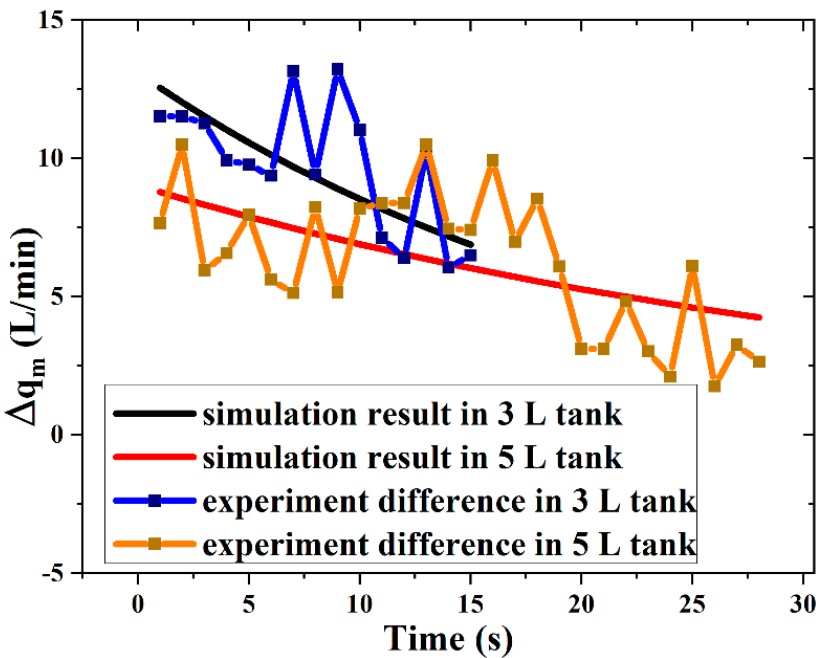

**Figure 10.** Comparison of experimental flow and calculated flow.

The goodness of fit, which confirms that the fitting effect is good, is shown in Table 4.

**Table 4.** Goodness of fit of flow index model.

| Tank | R-Square | RMSE |
|---|---|---|
| 5 L tank | 0.3359 | 2.11 |
| 3 L tank | 0.3861 | 2.504 |

The optimized results of the exponential model are displayed in Figure 11, which shows that the exponential model can eliminate the deviation of the quasi-isothermal cavity flow calculation.

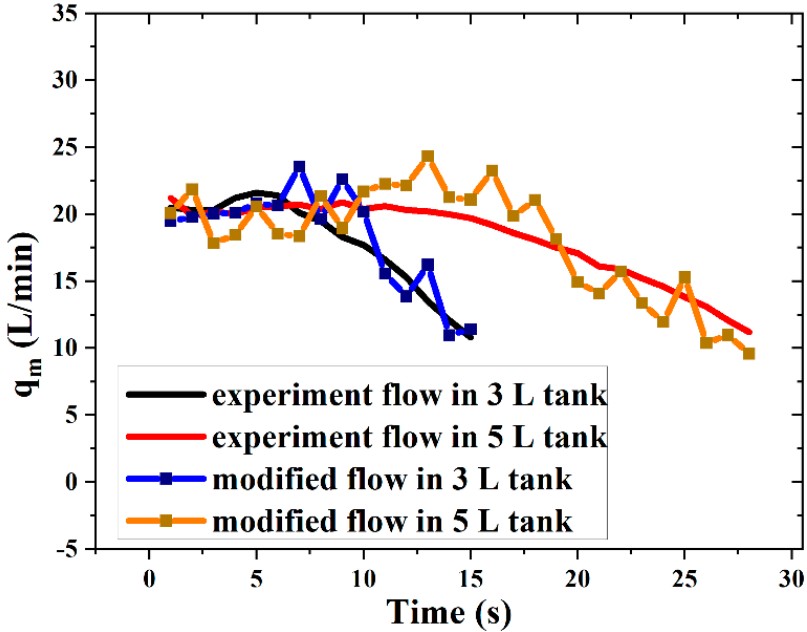

**Figure 11.** Fitting results of flow index model.

## 5. Conclusions

To solve the problems of complexity and inaccuracy in the existing standards for research on flow measurement without the use of a flowmeter, this paper presented a gas flow measurement method with temperature compensation for a quasi-isothermal cavity. In this paper, the measurement of the valve, the measurement of the gas tank charging flow, and the heat exchange in the gas tank were studied theoretically. The formula of the combination of the valve and the gas tank was derived, and the calculation of the overall heat transfer coefficient was obtained. At the same time, the identification of the temperature constant of the gas tank with copper wire was considered, and then the measurement method of the gas tank flow with temperature compensation was considered. Finally, the parameter identification of the exponential model of aerodynamic flow compensation was derived. During the experiment, the temperature increase after adding copper wire to a 5 L gas tank and 3 L gas tank was measured, and the temperature rise inhibition ratio was calculated. It was concluded that the filled copper wire could effectively inhibit the increase in temperature, which was helpful in building a quasi-isothermal environment. Then, in the experiment, the parameters of the exponential model were identified for the overall heat transfer coefficient and flow compensation, the error was calculated, and the conclusion that the compensation effect is better under these parameters was obtained.

**Author Contributions:** Data curation, Y.S. and L.L.; Methodology, J.C. and Q.Z.; Writing—original draft, L.L., Y.W. and Z.S. All authors have read and agreed to the published version of the manuscript.

**Funding:** This research was funded by Youth Fund of National Natural Science Foundation of China (No. 52105044), National Key R&D Program of China (No. 2021YFC0122502) and National Key R&D Program of China (No. 2019YFC0121702).

**Informed Consent Statement:** Informed consent was obtained from all subjects involved in the study.

**Data Availability Statement:** All data are available in this article.

**Acknowledgments:** Any support given was not covered by the author contribution or funding sections.

**Conflicts of Interest:** The authors declare no conflict of interest.

## Nomenclature

All the variables and parameters are described as follows.

| | |
|---|---|
| $p$ | Absolute pressure in certain situation (Pa) |
| $T$ | Thermodynamic temperature in the system (K) |
| $T_s$ | Initial temperature at when the air tank is not charging (K) |
| $T_r$ | Temperature in isothermal calculation (K) |
| $q_m$ | Flow rate of the system ($\mathrm{m^3 \cdot s^{-1}}$) |
| $\Delta q_m$ | Differential of flow rate of the system ($\mathrm{m^3 \cdot s^{-1}}$) |
| $V$ | Volume of the gas ($\mathrm{m^3}$) |
| $m$ | Mass of the gas (kg) |
| $R$ | Thermodynamic constant ($\mathrm{N \cdot m \cdot kg^{-1} \cdot k^{-1}}$) |
| $C_v$ | Specific heat at constant volume ($\mathrm{J \cdot kg^{-1} \cdot K^{-1}}$) |
| $h$ | Heat transfer coefficient per area ($\mathrm{Wm^{-2} \cdot K^{-1}}$) |
| $S_h$ | Heat transfer area ($\mathrm{m^2}$) |
| $K_1$ | Fitting parameter of overall heat transfer coefficient ($\mathrm{W \cdot K^{-1}}$) |
| $K_2$ | Fitting parameter of overall heat transfer coefficient ($\mathrm{s^{-1}}$) |
| $K_3$ | Fitting parameter of overall heat transfer coefficient ($\mathrm{W \cdot K^{-1}}$) |
| $K_4$ | Fitting parameter of overall heat transfer coefficient ($\mathrm{s^{-1}}$) |
| $K_5$ | Fitting parameter of pneumatic flow ($\mathrm{m^3 \cdot s^{-1}}$) |
| $K_6$ | Fitting parameter of pneumatic flow ($\mathrm{s^{-1}}$) |
| $\beta$ | Heating inhibition ratio |
| $t$ | Time of charging process (s) |

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
