# Peer review of "Gas Flow Measurement Method with Temperature Compensation for a Quasi-Isothermal Cavity"

_machines, doi:10.3390/machines10030178_

Round 1

Reviewer 1 Report

Referee comments on the paper Gas Flow Measurement Method with Temperature Compensation of Quasi Isothermal Cavity presented for publication in the Machines

This paper describes a method to determine the flow rate by measuring the pressure change in the process of gas tank inflation. This study also uses the method of temperature compensation to eliminate the temperature influence in the isothermal formula.

The work is quite interesting, but the introduction is written quite chaotically. Authors should revise citations with bibliography. Some items are missing. The literature record also needs to be improved. Selected errors are listed below

Line 59

Kagawa et al. – there is no such position in the bibliography with the first author. In position 21 there is an author with this name, but in the third position. We always quote the first author

Line 64

A similar situation, I do not understand the writing of position 11 in the bibliography

I also suggest placing the citation number directly on the name in the text, e.g. Oneyama [12,13]

Line 76

Authors write: Gao [16], should be Gao et al. [16]

Line 82

Chabane proposed…   [19], there is no such author under this pisition in the bibliography

Line 296

Authors write “according to the previous paper is..”, a citation should be given

Please also precisely define the purpose of the work

With these corrections, the work will become suitable for publication in Machines

Author Response

Dear Reviewer 1,

Thank you very much for your comments of the paper, and it has greatly guided our work. Your opinions have been carefully studied, and the article has been revised in details. This paper has been improved, and we considered that it is ready to be submitted again.

We have given our response to the reviews in details as uploaded attachments. Please download and browse. Thank you again for your valuable comments.

Reviewer 2 Report

This paper Gas Flow Measurement Method with Temperature Compensation of Quasi Isothermal Cavity is related to i.a. the important problem of the influence of temperature change on flow measurement. The article requires improvement and changes.

The most important remarks are:

  1. Please change/correct the abstract. It should be more precisely emphasized what is novelty, what are the advantages of this solution over the existing ones.
  2. Editorial errors (e.g. in line 91 – no dot, methods section – from the new side, very long sentences - sometimes difficult to understand (e.g. in line 59, 99, …)).
  3. Please comment on what the assumptions are based on - especially "Working under normal temperature and high pressure".
  4. Please change the Figure 1 caption.
  5. Please expand point 2.6 - for what purpose these equations were written?
  6. Please describe the measuring accuracy of the components used more precisely (3.1)
  7. One of the Figures (Figure 1 or Figure 2) is unnecessary - either a scheme or a figure of the experimental standup.
  8. Why such volumes of tanks and this solution - copper wire inserted into the air tank - were analyzed?
  9. In my opinion, the points on the graphs (Figure 3 and 4) should not be connected. Why is one data series a scatter chart and the other (with cooper wire) a line chart?
  10. Please improve the aesthetics of the figures.
  11. There is no description in the text for figures 7 and 8.
  12. The conclusions don’t emphasize the novelty of the analyzed subject or indicate the application area.
  13. Please change/correct the conclusion.

Author Response

Dear Reviewer 2,

Thank you very much for your comments of the paper, and it has greatly guided our work. Your opinions have been carefully studied, and the article has been revised in details. This paper has been improved, and we considered that it is ready to be submitted again.

We have given our response to the reviews in details as uploaded attachment. Please download and browse. Thank you again for your valuable comments.

Reviewer 3 Report

The manuscript requires a major revision. Please find my comments below.

Line 59: the author Kagawa is mentioned, however References point to Kawashima

Lines 63-64: was the sentence “However…wire.” intentionally put in this position?

Lines 71-89: the order of the references is incorrect

Line 77: there is no such author as Yang in the reference #18

Line 82: there is no such author as Chabane in the reference #19

In the Introduction the Authors mentioned about filling gas tank with the copper wire, however the idea is not clear. It should be clearly stated what solutions they found in the literature and which they applied in their work.

Line 112: why air compressor is supposed to be responsible for monitoring the pressure of output air?

Lines 119-120: copper wire filling is mentioned again without explanation; what does it mean that it is dense?

Line 122: Missing figure description.

Line 129: the mentioned ISO standard was neither quoted here nor before

Equations: all the variables and parameters used should be clearly described in the text and their units should be present

Line 145: equation (4) describes non-isothermal process

Line 146: shouldn’t be the volume of the chamber (tank) constant?

Line 233: what is “solar term”?

Subsection 3.1: were the 3L and 5L tanks connected to the system at the same time, or were they exchanged for each other during the measurements? Did the authors take into account the reduction in the volume of the tanks caused by filling them with copper wire?

Line 258: what do Authors mean by “the gas pressure is stable by a certain structure at the air source.”?

Figure 3a: why graphs don’t start with this same initial temperature?

Figures 3b and 4b: why graphs don’t start with this same initial pressure?

It seems if they started from this same level the differences “empty tank” vs “tank filled with copper wire” would be barely visible. Please explain this in detail.

Figures 3-6: if the measurement points are sparse presenting them enriches graphs and allows the readers to make their own conclusions

Lines 281-282: this sentence doesn’t describe Figure 5

Tables 1-3: missing units

References: many mistakes, for example: capital letters only (e.g. KUROSHITA), missing spaces (e.g. FluidsPart 1:General), presented authors’ names are their first names not surnames (e.g. Hubert, Daniel…), etc.

Vocabulary used is mostly correct, whereas grammar requires major correction. Therefore careful reading and correction by an English native speaking person is strongly advised.

Author Response

Dear Reviewer 3,

Thank you very much for your comments of the paper, and it has greatly guided our work. Your opinions have been carefully studied, and the article has been revised in details. This paper has been improved, and we considered that it is ready to be submitted again.

We have given our response to the reviews in details as Uploaded attachments. Please download and browse. Thank you again for your valuable comments.

Round 2

Reviewer 3 Report

I consider the corrected version of the manuscript is needed. Some language corrections are still needed. For example:

line 148: There is probably one “flow measurement” too much in the sentence “Flow measurement of gas tank charging flow measurement”

line 333: is “…it can see…”, probably should be “…it can be seen”

and some more.

Author Response

Thank you very much for your comments of the paper, and it has greatly guided our work. Your opinions have been carefully studied, and the article has been revised in details. This paper has been improved, and we considered that it is ready to be submitted again.

1, Text in Line 148 is changed to "2.3. Flow measurement of gas tank charging".

2, Text in Line 333 is changed to "Comparing theoretical and experimental results of overall heat transfer coefficient, it can be seen that the fitting effect is accurate".

3, The sentences in the manuscript were checked again.